# Diagnosing and Managing Uveitis Associated with Immune Checkpoint Inhibitors: A Review

**DOI:** 10.3390/diagnostics14030336

**Published:** 2024-02-04

**Authors:** Huixin Zhang, Lysa Houadj, Kevin Y. Wu, Simon D. Tran

**Affiliations:** 1Faculty of Medicine, Laval University, Quebec, QC G1V 0A6, Canada; huixin.zhang.1@ulaval.ca; 2Faculty of Medicine, University of Sherbrooke, Sherbrooke, QC J1G 2E8, Canada; lysa.houadj@usherbrooke.ca; 3Department of Surgery, Division of Ophthalmology, University of Sherbrooke, Sherbrooke, QC J1G 2E8, Canada; 4Faculty of Dental Medicine and Oral Health Sciences, McGill University, Montreal, QC H3A 1G1, Canada

**Keywords:** immune checkpoint inhibitor, uveitis, Vogt-Koyanagi-Harada, birdshot-like uveitis, immunotherapy, CTLA-4 inhibitors, PD-1 inhibitors, PD-L1 inhibitors, immune-related adverse events

## Abstract

This review aims to provide an understanding of the diagnostic and therapeutic challenges of uveitis associated with immune checkpoint inhibitors (ICI). In the wake of these molecules being increasingly employed as a treatment against different cancers, cases of uveitis post-ICI therapy have also been increasingly reported in the literature, warranting an extensive exploration of the clinical presentations, risk factors, and pathophysiological mechanisms of ICI-induced uveitis. This review further provides an understanding of the association between ICIs and uveitis, and assesses the efficacy of current diagnostic tools, underscoring the need for advanced techniques to enable early detection and accurate assessment. Further, it investigates the therapeutic strategies for ICI-related uveitis, weighing the benefits and limitations of existing treatment regimens, and discussing current challenges and emerging therapies in the context of their potential efficacy and side effects. Through an overview of the short-term and long-term outcomes, this article suggests recommendations and emphasizes the importance of multidisciplinary collaboration between ophthalmologists and oncologists. Finally, the review highlights promising avenues for future research and development in the field, potentially informing transformative approaches in the ocular assessment of patients under immunotherapy and the management of uveitis following ICI therapy.

## 1. Introduction

In the diverse arsenal of therapeutic tools against cancer, immune checkpoint inhibitors (ICI) have emerged in the last decade as a new beacon of hope. By inhibiting the immune response’s “OFF” signal, ICIs activate the body’s immune system to attack cancerous growths. Currently, eight immune checkpoint inhibitors have been approved by the FDA for their proven efficacy against multiple cancer types. Per their mechanism of action, ICIs produce a series of well-documented side effects secondary to the induction of immune activation commonly referred to as “immune-related adverse events” (IRAEs). These can affect any organ system, including the eye. Although rare, ocular IRAEs can have debilitating effects on patients’ quality of life and be sight-threatening. In the following sections, we will examine the literature and provide a comprehensive overview of drug-induced uveitis in the context of ICI therapy.

## 2. Overview of Immune Checkpoint Inhibitors (ICIs)

### 2.1. Brief Overview of Immune Checkpoint Inhibitors

#### 2.1.1. CTLA-4 Inhibitors

Cytotoxic T-lymphocyte-associated protein 4 (CTLA-4) is a transmembrane receptor preferentially expressed on regulatory T cells (Treg) and memory T cells [1]. Acting as a homodimer, it antagonizes CD28 signaling and suppresses the T cell response by competitively binding to CD80 and CD86, proteins expressed by antigen-presenting cells (APCs), as per Figure 1. Through a complex signal transduction pathway, CTLA-4 activation leads to reduced IL-2 secretion which limits T cell expansion and differentiation [2]. The CTLA-4 neutralizing antibody ipilimumab (Yervoy) was the first ICI approved by the FDA in 2011. Tables showing the indications, administration doses, and reported incidences of uveitis from CTLA-4 inhibitors phase I/II trials are found in Appendix A (Table A1).

#### 2.1.2. PD-1 Inhibitors

Programmed Cell Death Protein 1 (PD-1) is a cell surface protein expressed on natural killer (NK) cells, APCs (B cells, macrophages, dendritic cells), and activated T cells. Once a T-cell receptor (TCR)/CD28 interaction forms, PD-1 can be expressed, binds to programmed death ligand-1 (PD-L1) ligand, and mediates dephosphorylation of the TCR via the phosphatidylinositol 3-kinase (PI3K) pathway [3]. Thereby, PD-1/PD-L1 signaling antagonizes the positive feedback loop induced by TCR/CD28 binding, which ultimately halts cell cycle progression on both innate and adaptive immune responses [4]. In murine models lacking PD-1, a higher likelihood of developing autoimmune diseases has been observed [5,6]. Tables showing the indications, administration doses, and reported incidences of uveitis from PD-1 inhibitors phase I/II trials are found in Appendix A (Table A2).

#### 2.1.3. PD-L1 Inhibitors

PD-L1 is found on target tissues and binds to PD-1. Via a complex signaling pathway involving ZAP70 phosphorylation, PD-1/PD-L1 interaction inhibits T cell proliferation in the lymph node. PD-L1 expression is upregulated in various malignancies, particularly lung cancers. On the other hand, in autoimmune diseases like systemic lupus erythematosus, APCs fail to express PD-L1 [7]. Tables showing the indications, administration doses, and reported incidences of uveitis from PD-L1 inhibitors phase I/II trials are found in Appendix A (Table A3).

#### 2.1.4. LAG-3 Inhibitor

Lymphocyte activation gene-3 (LAG-3) found on activated T cells, inhibits T cell mitochondrial activity and is associated with CD4/CD8+ T cell exhaustion, making it a promising target for novel ICI therapies [8,9,10]. Studies have found that inhibiting LAG-3 and PD-1 led to increased cytotoxic T-cell activity and tumor response [11,12]. In March 2022, the FDA approved the first LAG-3 inhibitor, relatlimab, for use against unresectable or metastatic melanoma. The administration dose is 480 mg nivolumab plus 160 mg relatlimab intravenously over 30 min every 4 weeks [13]. No cases of relatlimab-linked uveitis have been reported as of the writing of this article.

### 2.2. Side Effects Other Than Ocular Side Effects

The incidence of IRAEs ranges between 64 and 72%, with high-grade IRAEs affecting up to 18–24% of patients undergoing ICI [14]. The timeline for the occurrence of IRAEs varies greatly. Though most IRAEs appear within 3–6 months of ICI initiation, delayed responses may take up to a year to appear, posing a challenge to diagnosis [15,16,17]. The incidence of IRAEs increases in a dose-dependent manner [14,18]. In a systematic review comparing ipilimumab 3 mg/kg and 10 mg/kg, authors found that the higher dosage group had a 3.10 greater chance of developing high-grade IRAEs [14]. This is not necessarily undesirable, however. In Downey et al. (2007), among 139 patients treated with ipilimumab for metastatic melanoma, all patients who experienced complete responses developed severe IRAEs, and the relationship between IRAEs and response was statistically significant [19]. The presence of IRAEs is positively correlated with increased survival. In a retrospective study of 133 patients, the overall survival of patients who developed IRAEs was thrice that of those who did not (37.8 months versus 10.1 months, respectively). The same study also found that patients who discontinued ICI had a mean survival time 30% shorter than those whose therapy was uninterrupted despite IRAEs [20]. Though a higher disease severity may be a confounding factor between increased mortality and ICI interruption, the clinical reflex to cease cancer therapy at once as side effects appear may need reconsideration. The non-ocular side effects of ICIs extend into a range of organ systems as illustrated in Table 1.

### 2.3. Ocular and Orbital Side Effects Other Than Uveitis

Ocular side effects associated with ICI therapy are generally rare and occur alongside other systemic IRAEs. Though their combined incidence is only around 1%–2.8%, early detection is necessary to prevent significant impacts on patient’s vision and quality of life [44,45]. Among phase I-III trials, dry eyes were the most reported IRAEs, with an incidence ranging from 1.2 to 24.2%, followed by uveitis at 0.3% to 6%. There were no reported cases of high-grade dry eyes, and only one small phase I study for combined nivolumab and ipilimumab in advanced melanoma reported more than one case of high-grade uveitis (two cases, or 4%) [44,46]. Most ocular IRAEs were reported in patients undergoing treatment for advanced or metastatic melanoma [44]. Though a wide range of ocular IRAE manifestations can be found in published case reports, as illustrated further in this section, interestingly, few diagnostic details are provided in larger drug trial studies. One study characteristically grouped all ocular symptoms under “dry eyes/blurred vision” [47]. Therefore, a close collaboration between ophthalmologists and oncologists is needed to better identify, prevent, and treat morbidity secondary to ICI while ensuring optimal continuation of ICI therapy.

Other than uveitis, some common ocular IRAEs (oIRAE) associated with the ocular system include:-Orbit;
○Giant cell arteritis [48];○Myasthenia gravis [49,50];○Inflammatory orbitopathy [51];○Cranial nerve 3/6/7 palsy [21,38];
-Anterior segment;
○**Dry eye** [52,53];○Corneal ulcer [51,54];
-Posterior segment
○Choroidal neovascular membrane/choroidal effusion [42,55,56];○Hypotony/macular edema [55,57,58,59,60,61];○Optic neuritis [62,63];○Retinal vasculitis [64,65];○Serous retinal detachment [55,66].


Note: common side effects (those affecting between 1 in 10 and 1 in 100 people) are **in bold** [14,20].

## 3. Pathogenesis of Uveitis Associated with ICIs

### 3.1. Association of ICIs with Uveitis

#### 3.1.1. Association with Combination Therapy

Risks of uveitis are significantly higher in combined therapy than in monotherapy. In lung cancer patients, the risk of developing uveitis during their treatment course is more than seven times higher amongst those receiving combined PD-L1/CTLA-4 inhibitors therapy than either ICI monotherapy [67].

#### 3.1.2. Associations with Non-Ocular IRAEs

An analysis of the WHO international pharmacovigilance database found that ICI-induced uveitis is most often reported in association with other cutaneous (10.4%), endocrine (9.0%), and neurologic (8.5%) IRAEs [68].

#### 3.1.3. Associations with Different Cancer Types

Uveitis was noted to be most reported alongside melanoma, which accounts for over 60% of reported cases, followed by lung cancer (20%) and renal cancer [29,38,54,61,66,69,70,71]. This has given rise to the hypothesis that advanced or metastatic melanoma may be directly involved in the pathogenesis of ICI-induced uveitis [72]. In a recent retrospective analysis of over 40,000 patients in the FDA Adverse Event Reporting System, the incidence of ICI-induced uveitis was established to be 0.45% among patients taking anti-PD-1/anti-PD-L1, anti-CTLA-4, or anti-PD-1/anti-CTLA-4 combination, all doses and durations combined, while this was almost three times higher, at 1.2%, in patients with melanoma [71]. Further, uveitis was almost five times more reported in melanoma than in lung cancer patients undergoing anti-PD-1/L1 therapy [71]. This result concurs with data from an analysis of the Kaiser Permanente Southern California health records, in which melanoma patients had an adjusted odds ratio of 6.45 times more likely to develop uveitis in the first year of ICI therapy compared to non-melanoma controls [73]. The pathogenesis of melanoma-induced uveitis is discussed in later sections.

#### 3.1.4. Association with Different ICI Types

A retrospective study by Braun et al. (2021) found that specific ICI classes were more strongly associated with uveitis than others. Ipilimumab (CTLA-4 inhibitor) was associated with the highest incidence rate ratio of uveitis (IRR = 30.5) among ICIs, exceeding even ipilimumab plus nivolumab combination therapy (IRR = 20.7) [73]. Compared to monotherapy via PD-1 and PD-L1 inhibitors alone, patients undergoing CTLA-4 inhibitor monotherapy developed anterior uveitis three times faster (all within 20 days in this latter cohort) [74]. The propensity for ipilimumab, a CTLA-4 inhibitor to induce marked IRAEs is confirmed in the literature [75,76]. On the other hand, VKH-like uveitis is significantly associated with PD-1/PD-L1 inhibitors, as discussed further in this article.

### 3.2. Mechanisms Triggering Uveitis

#### 3.2.1. Activation of the Complement Cascade 

Mechanisms triggering uveitis are not fully elucidated but there is some evidence that a type II hypersensitivity response may be involved. In a murine study of pituitary gland inflammation, site-specific deposition of C3 and C4 on prolactin- and TSH-secreting pituitary cells preferentially expressing CTLA-4 was observed, and anti-TSH and anti-PRL auto-antibodies were subsequently elevated in the serum of patients who developed hypophysitis after receiving ipilimumab [77]. According to this model, anti-CTLA-4 antibodies first bind to the cognate CTLA-4 antigen expressed on pituitary tissue. Subsequently, their Fc regions bind to C1q and activate the classical complement pathway, leading to C3, C3d, C4d recruitment, formation of membrane attack complexes, and cell lysis. These damaged tissues are then phagocytosed by antigen-presenting cells, leading to the production of anti-pituitary serum antibodies and further immune responses mounted against self-antigens. The type II hypersensitivity model may also explain why patients treated with ipilimumab, more than other ICI molecules, showed disproportionately higher incidences of IRAEs. Ipilimumab is an anti-CTLA-4 belonging to the IgG1 subclass, which offers preferential affinity for Fc-receptor binding, superseding other IgG subclasses as the best activator of antibody-dependent cell cytotoxicity response [78]. Comparatively, another commercialized CTLA-4 inhibitor, tremelimumab, an IgG2 subclass antibody, induced IRAEs at a much lower rate (1% versus 4%) in the same population [77]. It characteristically lacks affinity for the subclass of Fc receptors required to activate complements and failed to improve overall survival in clinical trials involving advanced melanoma, mesothelioma, and NSCLC [78,79].

#### 3.2.2. Role of Innate Immunity

Aside from the antibody-mediated complement activation cascade, other players in the innate immune system may also play a role in ICI-induced IRAE. In one case report of panuveitis induced by nivolumab, cytokine studies of ocular fluid during vitrectomy revealed elevated levels of G-CSF and CXCL10 [80]. G-CSF is central to the production of neutrophils and other granulocytes; and CXCL10 is a chemokine for natural killer, dendritic, and macrophage cells, aside from T/B cells [81]. Though previous treatment with corticosteroids may have altered this result, this cytokine profile suggests that neutrophils, natural killer cells, and macrophages may also be involved in the intraocular inflammatory response [80].

#### 3.2.3. Role of Humoral and Cell-Mediated Autoimmunity 

The pathogenesis of melanoma-induced uveitis may be explained by an inadequate cross-reaction between melanoma and ocular antigens. A study of dermatological IRAEs pointed to increased melanocytic differentiation marker Melan-A-positive CD8+ T cells infiltration at the rash sites and in the serum. According to the authors, anti-CLTA-4 therapy releases the brake on self-tolerance, allowing auto-reactive Melan-A-specific T cells to expand and exert damage on the tumor site [82]. Indeed, in autoimmune uveitis, loss of tolerance to various self-proteins, such as melanin, retinal arrestin, and recoverin is reported [83,84,85]. In one study, where the authors induced anterior uveitis by sensitizing with bovine uveal melanin, the authors determined that the response was probably cell-mediated as serum transfer did not induce uveitis in the host [84]. In non-ICI-induced Vogt–Koyanagi–Harada (VKH) disease, T-cell clones isolated from five patients demonstrated reactivity against tyrosinase and tyrosinase-related protein 1, suggesting these self-antigens may be involved in the pathogenesis of VKH [86]. Together, these studies suggest a humoral and cell-mediated hyperactive immune response may be involved in the loss of self-tolerance behind ICI-induced uveitis. The antigen specificity and activation mechanism of these T cells still need to be elucidated.

On the other hand, it is worth mentioning that vision loss in neoplasia can be caused by paraneoplastic syndromes linked to self-antibodies, with or without ICI therapy. For example, melanoma-associated retinopathy (MAR) is believed to arise from antiretinal antibodies in response to antigens expressed on both melanocytes and retinal bipolar cells [87,88]. There are reports of MAR worsening with ICI [89,90,91]. With treatment, melanoma shrinks and disappears, and MAR symptoms improve [92,93]. This evidence supports the hypothesis that self-antibodies play a role in oIRAEs.

#### 3.2.4. Loss of Immune Privilege 

The intraocular space is an immune-privileged site, protected by local and systemic factors. One such factor is the constitutive expression of PD-L1 in the cornea and retinal pigmental epithelium [94,95]. Unsurprisingly, PD-1/PD-L1 inhibitors disrupt this fine balance and lead to ocular inflammation. In the 2020 retrospective analysis of the FDA Adverse Event Reporting System database, among more than 92,000 IRAES reported, it was found that anti-PD-1/PD-L1 therapy was significantly associated with uveitis as oIRAE. Furthermore, the risk of uveitis under these therapies has a strong differential association with some cancers such as melanoma, suggesting a combination of factors leads to uveitis development [71].

### 3.3. Genetic and Environmental Factors

#### 3.3.1. Sex and Extracellular Hormonal Environment

Risk factors associated with the development of ICI-related adverse events can be either genetic or environmental. It’s been established that estrogen levels favor T-cell differentiation into CD4+ helper as opposed to CD8+ cytotoxic cells and that females exhibit more of a humoral immune response [96]. Therefore, males may have more CD8+ T cells leading to targeted local tumor responses, whereas females may experience more tissue damage and IRAEs owing to higher excess free immune complexes. However, retrospective case studies failed to show this difference. In a cohort of 231 NSCLC patients receiving anti-PD-1 therapy, the female sex was associated with higher risks of IRAEs (OR = 1.12) [97]. However, when controlled for other factors, sex and age no longer played a significant role [67]. Furthermore, sex was not associated with the severity and hospitalization rate of IRAEs [97,98]. In studies of both melanoma and NSCLC cohorts, men and women regardless of age had equal rates of grade ≥ 3 IRAEs [97]. 

#### 3.3.2. HLA Predisposition

One’s genetic makeup plays a role in IRAE susceptibility. The human leukocyte antigen (HLA) gene, encoded on chromosome 6 has many variants in the human population, and some have been preferentially associated with autoimmune diseases [99]. Case reports described links between ICI-induced VKH disease-like uveitis and HLA-DRB1*04:05 and/or HLA-DRB1*04:10 [69,70,100,101,102,103,104]. The allele is strongly associated with autoimmune diseases and 35% of VKH patients in the general population, with one study finding 100% specificity for VKH in a study of 63 Japanese patients [105,106,107]. Various single nuclear polymorphisms (SNPs) have also been associated with IRAEs. Some differentially modulate PD-1/PD-L1 expression and have been linked to more than a 40% increase in progression-free survival from nivolumab, a PD-L1 inhibitor [108]. Therefore, genetic variations may explain one’s response intensity to ICI—with stronger responses potentially associated with drug-induced uveitis.

#### 3.3.3. Role of the Microbiome

In recent decades, the microbiome has emerged as a novel focal point of interest in immune regulation research. Evidence shows that gut dysbiosis can modulate response to immune checkpoint inhibitors, and the latter—once thought to be encoded in one’s DNA, is altered following fecal transplants [109]. The presence of distinct species of *Bacteroides*, for example, contributes to the differentiation of Treg and is needed in mice to mount an adequate response to CTLA-4 inhibitors [110,111]. Intriguingly, mice harboring these bacteria were protected against ICI-induced colitis, indicating IRAEs may be modulated by the composition of the gut microbiota as well [111].

## 4. Manifestations of Uveitis Associated with ICIs

Uveitis is one of the most common ocular IRAEs associated with ICI—one study calculated uveitis made up almost 70% of all ocular IRAEs, of which more than 80% were bilateral at diagnosis [72]. In a retrospective cohort study of the Medicare database, patients undergoing cancer treatment with ICI therapy were twice more likely to develop non-infectious uveitis compared to their non-ICI counterparts [112]. The CTCAE is used to grade uveitis resulting from adverse effects of cancer treatments (Table 2) [113].

### 4.1. Anterior, Intermediate, Posterior, Panuveitis 

The classification of uveitis is defined according to the International Uveitis Study Group system, which is the classification used in the present paper [114]. Four types of uveitis are found in association with ICIs. 

(1)Anterior uveitis, anterior cyclitis, iritis, or iridocyclitis, affects the anterior chamber of the eye. It is characterized by red-eye photophobia, acute blurred vision, and pain/discomfort. Typical signs of anterior chamber inflammation include conjunctival congestion, corneal edema, keratic precipitates, presence of cells and flares, and iris synechiae [115]. The number of cells may vary from trace to hypopyon formation [116];(2)Intermediate uveitis, hyalitis, posterior cyclitis, or pars planitis, refers to inflammation of the vitreous body. Symptoms include blurred vision and floaters;(3)Posterior uveitis, retinitis, neuroretinitis, chorioretinitis, or choroiditis, refers to inflammation of the retina, retinal vessels, and choroid. Principal symptoms consist of flashing lights, floaters, or vision loss. Slit lamp examination may include cells in the posterior chamber and vitreous haze. Fundus exam reveals retinal fluid or chorioretinal lesions [117,118];(4)Panuveitis occurs when inflammation is localized to more than one of the above compartments [119].

The working group description of uveitis also defines the course of uveitis as acute, recurrent, and chronic [120].

A majority of anterior uveitis are reported in the literature [64,69,121,122,123,124,125,126]. In a literature review of 241 eyes that were diagnosed with uveitis following ICI therapy, anterior uveitis was the most common type of uveitis (37.7%), followed by panuveitis (34.0%), posterior uveitis (25.7%), and finally intermediate uveitis (1%). This distribution was similar regardless of the ICI used, but patients under a combination of ICIs demonstrated a higher rate of anterior uveitis [127]. Martens et al. (2023) also described a strong discrepancy between anterior and panuveitis, and intermediate uveitis. Among 134 reported cases, they found ICI-induced anterior uveitis accounted for 45%, panuveitis 42% (of which 22% were VKH-like), posterior 4%, and intermediate 6% [128]. A review of uveitis manifestations documented in phase I-III trials showed comparable results [71]. Whether the preferential induction of anterior and panuveitis rather than intermediate uveitis is linked to the pathogenesis of ICI-associated uveitis remains to be explored. Atezolizumab (anti-PD-L1) had a heavy tendency towards posterior uveitis, with a percentage of posterior uveitis cases three-fold higher than any other ICIs (80% vs. 23.7%), while no cases of anterior uveitis were reported. The percentage of developing panuveitis was doubled for patients under ipilimumab (anti-CTLA-4) and nivolumab (anti-PD-1) monotherapies compared to other molecules [127].

The severity of visual acuity disturbance can vary, ranging from 20/20 vision to light perception, but the median visual acuity tends to be around 20/40 [127]. As expected, median presenting visual acuity tends to be better in patients with anterior uveitis (20/30) and worse in patients with intermediate or panuveitis (20/40–20/50) [129]. The distribution of uveitis severity varies—most were of grade 2 (15–37.7–65%) and 3 (35–70%), with few cases of grade 4 ICI-induced uveitis (<16.5%) [127,130]. 

### 4.2. Vogt–Koyanagi–Harada-like Uveitis 

Vogt–Koyanagi–Harada (VKH) syndrome is a rare form of bilateral panuveitis that occurs when inflammation is triggered against melanocyte-containing choroidal tissue and spreads into neighbouring structures [131]. It is characterized by granulomatous anterior or posterior uveitis, with thickening of the choroid, papillitis, and serous retinal detachment, possibly associated with central nervous system, auditory, or dermatological manifestations. Depigmentation of the fundus and limbus several weeks after the episode may leave characteristic sunset-like lesions (sunset glow fundus) as shown in Figure 2 [132].

VKH-like uveitis in the context of ICI therapy is occasionally reported. A study of individual case safety reports from the WHO international pharmacovigilance database found that ICIs and protein kinase inhibitors were the only drugs associated with drug-induced VKH-like uveitis [68]. Our literature search yielded 24 retrospective studies, case reports, and case series analyzing VKH-like uveitis in the context of ICI therapy, totalling 41 patients [68,69,70,100,101,102,103,134,135,136,137,138,139,140,141,142,143,144,145,146,147,148,149,150]. Details of the included study data are found in Appendix B (Table A4). Notably, most were women with a mean age of 68 years old, being treated for melanoma (59%) with a PD-1 or PD-L1 inhibitor (76%). There is a high incidence of HLA-DR4 or HLA-DRB1 positivity (37%), and the actual value may be higher than reported.

### 4.3. Birdshot-like Posterior Uveitis 

Birdshot uveitis (BU), also called birdshot chorioretinopathy, is a form of chronic bilateral posterior uveitis that, in the general population, mainly affects Caucasian males in their sixtieth decade of life [151]. It has a strong association with the HLA-A29 allele, and the pathogenesis of disease apparition is linked to endoplasmic reticulum dysfunction leading to an unknown antigen stimulating cytotoxic T cell activation [152,153]. On slit lamp and fundus examination, characteristic choroidal “birdshot lesions” (Figure 3) located inferiorly or nasally to the optic disk, low-grade anterior chamber and vitreous inflammation are required criteria for diagnosis [154]. On OCT images, retinal vasculitis and cystoid macular edema may also be observed [154]. BU is thought to make up 6–8% of all posterior uveitis cases [155].

Few reports of birdshot-like uveitis induced by ICI therapy exist: our literature review yielded two case reports [157,158]. In the first instance, preexisting BU relapsed during pembrolizumab therapy—a 56-year-old Caucasian female, in the context of a new NSCLC diagnosis, had to discontinue her existing immunosuppressive therapy [158]. Eight months later, her ophthalmologist documented an increase in cystoid macular edema. However, it is unclear here whether her exacerbation was due to discontinuation of immunosuppressive therapy, introduction of ICI, or both.

On the other hand, Acaba-Berrocal et al., 2018 described the case of a mid-60s female with metastasizing melanoma presenting with de novo retinal vasculitis, macular edema, and bilateral multifocal uniform choroidal lesions, worse in the left eye. Although vitreous inflammation was absent, the diagnosis of birdshot-like uveitis was retained as the patient had characteristic symptoms of blurred vision, nyctalopia, and floaters [157]. She was found to be HLA-A29 negative, so her cell-mediated autoimmune reaction mimicking BU was likely provoked by pembrolizumab. This case raises the possibility that some BU may be reported as posterior uveitis as both share similar clinical and imaging findings and vice versa. Further studies are needed to clarify the link between posterior uveitis and birdshot-like uveitis induced by ICI therapy.

### 4.4. Symmetry and Laterality

ICI-induced uveitis is predominantly bilateral. In one study of 124 patients, more than 90% had bilateral uveitis, of which almost 95% had concordant uveitis classifications in both eyes [127]. This percentage is concordant with another cohort study which found 88.5% of patients had bilateral uveitis at diagnosis [130].

### 4.5. Complications of Uveitis

Hypotony may occur as a complication of uveitis—this can lead to irreparable vision loss [55,57,58,59,61]. Reported causes of hypotony-related vision loss include maculopathy, choroidal, and retinal detachment [55,58]. Descemet membrane folds may occur as a result of anterior uveitis [159]. Epiretinal membranes have been described following ICI-induced uveitis as a result of intraocular inflammation [160,161]. Macular edema can rise in high-grade uveitis, causing metamorphopsia and vision loss [160,162].

## 5. VIII. Epidemiology and Risk Factors of Uveitis Associated with ICIs 

### 5.1. Age

In a small retrospective study involving mostly melanoma patients, age and sex were not found to be significant risk factors, although premenopausal women (under age 52) were more likely to develop higher severity IRAEs and have their ICI treatment discontinued as a result [97,163]. Age does not seem to be a risk factor in ICI-associated uveitis specifically [67,97]. 

### 5.2. Sex

Though females are more likely to develop IRAE, it does not seem to play a significant role in ICI-induced uveitis frequency and severity among melanoma and lung cancer patients as previously described [97].

### 5.3. Ethnicity

Race does not seem to play a role in ICI-induced uveitis among lung cancer patients [67]. Birdshot-like uveitis preferentially affects older Caucasians. VKH-like uveitis is often associated with the Japanese population, which has a higher incidence of HLA-DR4 and HLA-DRB1 [164].

### 5.4. Prior Ocular History

Prior episodes of uveitis or other ocular inflammatory conditions are a risk factor [74]. Specifically, 38.9% of patients with a known history of anterior uveitis experienced flare-ups under ICI, whereas those with recorded past intermediate, posterior, or panuveitis had a combined recurrence rate of 51.1% [73]. This rate is significantly higher than the expected rate of uveitis (3.6%) in the same uveitis-free cohort. History of ocular trauma or surgery is also a risk factor, associated significantly with ocular inflammation in an analysis of 40 patients who experienced ocular IRAEs [165].

### 5.5. Past Medical History

A past medical history of autoimmune disease or renal failure are risk factor for developing ICI-related uveitis. However, the use of corticosteroids before starting ICI treatment was found to be protective in this patient population with high percentages of melanoma [163].

### 5.6. Cancer Type

As previously mentioned, melanoma patients were two to three times more likely to develop ICI-related uveitis compared to other cancer types [166]. After melanoma, the second cancer type most significantly associated with uveitis is lung cancer, but its risk is much lower [67].

### 5.7. ICI Class

Various studies analyzed the contribution of ICI class on the risk of drug-related uveitis. A study analyzing the FDA Adverse Events Reporting System found ipilimumab and nivolumab combination therapy to have the strongest association with uveitis, along with others which showed CTLA-4 inhibitors were associated with higher risks of anterior uveitis as related in Section 4. On the other hand, a study of 40 ICI-induced uveitis cases found pembrolizumab therapy was more highly associated with intraocular inflammation, although the analysis did not differentiate between anterior and posterior uveitis and included ophthalmoplegia [165]. PD-1/PD-L1 inhibitors seem to be differentially associated with posterior uveitis as related above. Regardless, ICI therapy is a risk factor for drug-related uveitis, with an incidence of 0.3% to 0.4% which is significantly higher than other drug classes [45,112].

## 6. Diagnostic Tools

### 6.1. Diagnostic Criteria

Drug-related uveitis is often diagnosed clinically in the context of a negative infectious and autoimmune workup and temporal relation with the administration of ICI. Though more applicable in scientific research than routine clinical settings, the Naranjo criteria is a useful tool to assist clinicians in weighing the probability that ICI therapy caused the observed adverse reaction (Table 3) [167,168,169].

### 6.2. Laboratory Investigations

Patients presenting with signs of uveitis during ICI treatment must undergo rigorous laboratory investigations to rule out infectious and autoimmune causes. The current section describes a complete uveitis laboratory workup. 

Serologic testing for antinuclear antibodies (ANA), antineutrophil cytoplasmic antibodies (ANCA), dsDNA antibodies, and rheumatic factors is needed to exclude autoimmune diseases. Serum angiotensin-converting enzyme (ACE) is a reliable marker of sarcoidosis, which can manifest with uveitis in 30–70% of cases [170]. A genetic workup for autoimmune HLA susceptibility can be ordered. HLA-B27 is particularly associated with spondyloarthropathy and other autoimmune disorders, which can manifest in the eye as acute anterior uveitis [126,171]. As previously mentioned, HLA-A29 supports the diagnosis of birdshot chorioretinopathy, while HLA-DR4/DRB1 is associated with VKH-like uveitis. 

Ocular Epstein–Barr virus (EBV), toxoplasmosis, herpes simplex, syphilis, Lyme disease, and tuberculosis must be ruled out as infectious causes. Serologies, fluorescent treponemal antibody absorption test (FTA-ABS), *Treponema pallidum* particle agglutination (TP-PA), rapid plasma regain (RPR), and/or QuantiFERON-TB tests can thereby be obtained [123,172].

The workup is completed with a complete blood count, blood urea nitrogen, serum creatinine, liver function test, C-reactive protein/erythrocyte sedimentation rate, and urinalysis [123]. A chest *x*-ray is indicated when suspicion of tuberculosis is high.

### 6.3. Imaging Investigations

Imaging studies are essential in the ophthalmologist’s toolkit, assisting physicians in detecting vision-threatening illnesses. In a patient presenting with signs of uveitis in the context of ICI therapy, various imaging modalities may be prescribed according to symptoms and slit lamp findings. 

Symptoms of blurred vision or vision loss warrant an optical computed tomography (OCT). OCT allows visualization of intra-retinal abnormalities (fluids or folds) in the setting of a posterior uveitis [117,173]. It can also demonstrate complications of uveitis such as macular edema, epiretinal membrane, intraretinal bleeding, and optic nerve pathology [61,174]. When it is difficult to visualize the anterior chamber due to inflammation or corneal opacity, anterior segment OCT may be useful [175]. Uveitis grading based on hyperreflective spots from the anterior segment OCT provides an objective assessment of ocular inflammation that correlates strongly with clinical examinations [176,177,178]. Fibrin, corneal edema, inflammatory cells, and keratic precipitates are easily visualized using this imaging modality [175]. Newer innovative OCT imaging technologies, such as dye-free OCT angiography, may also provide useful information if readily available in the clinic [179]. The latter measures parafoveal capillary density, which is significantly lowered in eyes with uveitis regardless of macular edema [180,181]. It is particularly useful in detecting choroidal neovascularization that appears as a “lacy vascular network” on a grey background of a normally avascular outer retina [182].

Follow-up of retinal hemorrhages may require serial fundus photographs [116]. Color fundus photography is useful for the visualization of the retina. Signs of optic disk pathology in the fundus exam warrant a fundus fluorescein angiography (FFA). Fluorescein or indocyanine green angiographies are helpful to visualize vascular changes in posterior uveitis, birdshot chorioretinopathy, and VKH-like uveitis [66]. They reveal vascular leakage evidence of vasculitis, neovascularization, retinal artery or vein occlusion, and the like [172,183].

B-scan ultrasonography follow-up is routine in patients treated for choroidal melanoma and is useful for characterizing ICI efficacy over time. Skin melanoma and other cancers can rarely metastasize to the eye and present as decreased vision and inflammation [184]. B-scan ultrasound can differentiate intravitreal tumors and is a useful adjuvant technique for visualizing posterior segment pathology and complications. Specifically, it can detect subretinal fluids and choroidal thickening (normal thickness is around 1.1 mm) such as found in VKH uveitis [185]. Hypoechoic images or opacities in the vitreous point to intermediate uveitis or vitritis. Retinal detachment, a complication of uveitis, is also readily detected [186].

In patients presenting with acute visual field loss who are undergoing ICI treatments for advanced or metastatic cancer, a brain and orbit MRI may be warranted if vision loss due to metastasis is clinically suspected [187].

### 6.4. Specialized Diagnostic Tests

Both paraneoplastic-associated retinopathy and ocular IRAE induced by ICI are rare manifestations that may present with vision loss, visual field defects, and color vision deficiency [188,189]. To differentiate them, clinical and dilated fundus exams may be unrevelatory, especially in the early stage of the disease. An electroretinogram (ERG) can reveal classical signs of melanoma-associated retinopathy (MAR) by measuring the electrical activity of different retinal cells in response to light [189]. Specifically, absent or prominently reduced b-waves—attesting to “on” bipolar cell dysfunction due to destruction by self-antibodies—in the presence of normal a-waves (normal photoreceptor functions) are hallmark signs of MAR [188,190]. Interestingly, ERG changes in MAR normalize with ICI treatment, probably as a result of tumor regression [92]. In optic nerve pathologies such as optic nerve edema, delayed optic nerve conduction is expected along with signs of rod and cone dysfunction: increased latency and depressed a-wave/b-wave are observed [160].

Built on the principle of ERG, a newer version called a multifocal electroretinogram (mfERG) allows simultaneous measurement of a large area of the retina to light stimulus, generating a 3D, color-coded image optimized for visual interpretation [191]. This innovative technology has shown promise in detecting early signs of photoreceptor damage before the appearance of OCT changes such as macular edema in posterior uveitis [138]. 

Commercially available retinal antibody panels are rarely accessible in a day-to-day setting. In a study of 14 serum samples from patients without autoimmune retinopathy, 93% showed the presence of antiretinal antibodies [192]. Aside from recoverin, which is specific for carcinoma-associated retinopathy, routine retinal antibody testing for detection of autoimmune retinopathy is not recommended due to its low specificity [193,194].

In vivo confocal microscopy (IVCM) is an imaging modality used in scientific research that has emerged as an increasingly popular tool in ocular diseases. Though not routine in most clinics, it can provide interesting data on the dynamic processes of the cornea in real-time. Its capacity to image through opaque corneal tissue is an asset [195]. In uveitis, IVCM is useful for detecting stromal edema, keratic precipitates, and blebs of active inflammation appearing as lacunae in the endothelium [196]. Some small studies suggest IVCM may be useful in distinguishing infectious and non-infectious uveitis based on the sizes and shapes of keratic precipitates observed under the IVCM [197]. Though clinical applications for these findings may be superseded by their practicability, they offer a glimpse into the wealth of imaging modalities available to a clinician’s fingertips in the workup of uveitis.

## 7. Management of Uveitis Associated with ICIs

### Early Recognition and Collaboration between Oncologists and Ophthalmologists

Ocular immune adverse events tend to occur sooner than other adverse events [128,198]. Ocular adverse events can represent the first manifestation of systemic auto-immune reactions triggered by ICIs. The typical onset of uveitis is widely variable, ranging from 5 days to 5 years [128]. In a review of ocular adverse events combining 290 patients treated with ICIs for different cancers, the majority being lung cancer and melanoma, the earliest and latest cases of uveitis were discovered at 2 weeks and 2 years, respectively, after initiation of immunotherapy [128]. Most cases in the literature are diagnosed within the first 6 months of ICI initiation [72,127,199,200]. In their review of 126 cases Dow et al. noted that 83.6% to 91.67% of patients with uveitis were diagnosed within 6 months of treatment with ICIs [127]. In this study, patients developed uveitis at a median time of nine weeks from ICI start [127]. In Sun et al.’s retrospective study, most of the 51 cases of uveitis studied occurred during the first 20 days [74]. Chaudot et al. presented 22 cases of ICI-induced uveitis and most were diagnosed within the first 20 weeks of starting immunotherapy [72]. Many other studies yielded similar results with median occurrence time estimated at 76 days, 41 days and 70 days [68,76,198]. Zhou et al. reported that the average time of onset of uveitis in patients treated with ICIs for lung cancer was 32.22 days, that is, significantly shorter than the average time of onset of other ocular immune-related adverse events of ICI [201]. In this same study, most cases of uveitis occurred within the first ten days, regardless of age, sex, and ethnicity. These results suggest that uveitis can happen quite early in the treatment course and, more importantly, that it can be the first sign of ocular immune-related adverse events. Early detection is therefore crucial, not only to initiate prompt treatment but to help ophthalmologists recognize patients at higher risk of developing other types of IRAEs further in the treatment course. 

## 8. Treatment Options

### 8.1. Topical Steroids and Systemic Corticosteroids

Management of ICI-associated uveitis is necessary to avoid sight-threatening complications. The cornerstone of ICI-related uveitis treatment is immunosuppression through corticosteroids. The preferred administration route depends on several factors including the type and grade of uveitis. Clinical judgment is necessary to optimize treatment while minimizing side effects and impact on tumor progression. 

When possible, local corticosteroids are preferred over systemic corticosteroids since the latter have a greater immunosuppressive effect, and their impact on immunotherapy efficiency and tumor progression is not completely understood. Large doses and chronic administration of corticosteroids are generally not recommended, not only because they dampen immunotherapy efficiency but also carry a significant risk of ocular and systemic complications. Most studies agree that mild anterior uveitis can be effectively managed solely with local steroids as a first-line treatment [116,166,201]. In his study, Zhou et al. reported that almost all their 15 cases of ICI-associated uveitis were managed with topical or periocular corticosteroids and that systemic steroids were used as second-line treatment for disease refractory to topical steroid [201]. In Dow et al., 72.1% of patients were treated with a combination of topical steroids and other treatment modalities, but 36.9% of patients, typically those presenting with anterior uveitis, were under topical steroid therapy only. Topical treatment included prednisolone acetate 1% drops four times daily and difluprednate 0.05% drops four times daily [127]. In a retrospective chart review conducted at three tertiary ophthalmology clinics in the United States, among eleven patients diagnosed with ocular adverse events, nine had uveitis. Six of them received topical corticosteroids and three systemic corticosteroids. For most patients, this treatment controlled the ocular inflammatory response and ICI was pursued. Ocular IRAEs appeared to be easily controlled by local or systemic corticosteroids and did not require routine discontinuation of ICPIs [144]. Similar results were described by Parikh et al. in a case series of five patients diagnosed with acute anterior or intermediate uveitis of grade I to III following therapy with Nivolumab, Crizotinib, Durvalumab, or Daratumumab. All five cases were efficiently controlled with topical prednisolone acetate 1% tapered to the minimal daily dose necessary to pursue ICI therapy. No patients required discontinuation of the immunotherapy [116]. 

Treatment recommendations based on the CTCAE grading scale were published by the American Society of Clinical Oncology (ASCO), the Society for Immunotherapy of Cancer (SITC), and the National Comprehensive Cancer Network (NCCN) [127]. According to these recommendations (Table 4), low-grade uveitis can be managed with local corticosteroids. For moderate uveitis (CTCAE grade 2), discontinuation of ICI is recommended and local and/or systemic steroids are used. For more severe uveitis (CTCAE grade 3–4), high-dose systemic steroids are indicated. For these cases, suspension of immunotherapy is recommended and initiation of biologic immunosuppressants can be necessary if refractory. Re-introduction of immunotherapy is not recommended after a grade 4 uveitis episode. 

Chaudot et al. found that more than half (*n* = 36) of the 62 anterior uveitis cases reported were treated solely with local steroids. Additionally, 11 of the 26 cases of panuveitis were also managed with local steroids only [72]. Local steroid administration routes for ICI-induced uveitis are not limited to eye drops. Other types of local treatment such as subconjunctival, peribulbar, retrobulbar steroids, intravitreal dexamethasone implant, and triamcinolone periocular space injection are also reported [128]. In Dow et al., these alternative local steroid treatments were used in 20.7% of cases, with sub-Tenon triamcinolone and intravitreal triamcinolone injections being the most common delivery methods. On the other hand, 53.2% of patients were treated with systemic steroids, most commonly with oral prednisone or less commonly with intravenous methylprednisolone [127]. In a retrospective case series of eight patients treated with different cancer immunotherapies, including 4 treated with ICIs, local corticosteroids allowed good control of ocular inflammatory activity. In patients who continued their ICI therapy, sub-Tenon triamcinolone injections effectively treated uveitis and avoided ICI discontinuation and systemic corticosteroids [202]. 

Systemic corticosteroids were proven effective in the treatment of ICI-associated uveitis by numerous studies [100,202,203]. Typically, when there is posterior involvement, systemic corticosteroids are initiated as a first-line treatment with a possible association of immunosuppressors, immunomodulators, or biologics. A major issue with systemic corticosteroids is their counteractive effect on immunotherapy. While ICIs potentialize the body’s immune response against neoplastic cells, corticosteroids have an immunosuppressive effect, decreasing immune function and promoting infections and cancer development. Interestingly, systemic steroids were shown to decrease tumor response to anti-PD1 and anti-PDL-1 ICIs for lung cancer when the daily dose exceeded 10 mg/kg, but seem to not affect the efficacy of anti-CTLA-4 [204].

### 8.2. Immunosuppressor, Immunomodulator, and Biologics

As seen previously in the recommendations for the treatment of ICI-associated uveitis, immunosuppressors and biologics can be indicated in uveitis of grade III to IV and in patients with refractory disease. In some patients with ocular or systemic comorbidities, systemic corticosteroids are not recommended, which limits the treatment choice. Antimetabolites such as azathioprine, mycophenolate mofetil, cyclosporine C, and methotrexate seem to yield good results [68,158]. These molecules are already used in the treatment of chronic ocular inflammatory diseases such as juvenile idiopathic arthritis-associated uveitis. No studies are comparing the safety profiles of immunomodulators for ICI-induced uveitis specifically, but there exist data for other types of uveitis, namely juvenile idiopathic arthritis-associated uveitis. For this specific type of uveitis, methotrexate is generally the first choice since it carries a good efficacy profile and was not proven to increase the risk of cancer [205]. However, this condition is more often seen in the pediatric population, and these results may not apply to the adult population. Alternatively, biologics such as tumor necrosis alpha (TNF-α) inhibitors, including infliximab and adalimumab, can be used in rare cases when antimetabolites are not sufficient to control inflammation, and steroids are not suitable or sufficient. There are reports of cases of VKH uveitis for which the introduction of adalimumab allowed a significant reduction in the daily prednisolone dose (from 16.9 ± 7.9 mg to 6.3 ± 3.1 mg) [206]. However, these patients had to be closely monitored for cancer recurrence. Indeed, the use of anti-TNF-α in patients undergoing immunotherapy for cancer has not been extensively studied yet; therefore, its safety and effect on tumor progression are unknown [128]. Chronic immunosuppressive therapy has previously been associated with higher risks of developing infections and cancers [207,208] In a single-center retrospective cohort study of 190 adults with inflammatory eye disease treated with long-term systemic corticosteroids and immunosuppressive therapy, 25 malignancies were observed in 17 patients. The incidence of cancer was higher, evaluated at 2.10 per 100 person-year in the group treated with immunosuppressants, compared to 0.43 per 100 person-year in the systemic corticosteroid group [207,208].

Finally, immunomodulatory treatments like plasma exchange and intravenous immunoglobulins are also possible alternatives for patients with contraindications to corticosteroids, with tumor progression under steroid therapy or with severe or refractory disease, but were not often used in ICI-induced uveitis cases reported in the literature [209].

### 8.3. ICI Discontinuation

According to the previously presented guidelines, the decision to discontinue immunotherapy in a patient should be based on the severity of ocular inflammation. ICIs should be discontinued if uveitis is classified as grade 2 or higher and until inflammation decreases to grade 1. If ICIs are stopped, ASCO guidelines recommend waiting until systemic steroids are tapered to a daily dose of 10 mg oral or less before re-initiation of ICIs. Numerous case reports describe a tolerant approach for the majority of grade I and even grade II uveitis, as they can be managed without stopping the ICI. When uveitis is recognized promptly, systemic treatment with corticosteroids can be initiated rapidly and vision can be preserved without cessation of immunotherapy in more than 90% of cases [210]. 

Some case reports even describe cases of higher-grade uveitis where a decision was made to pursue immunotherapy. Tugan et al. published a case report of a 60-year-old patient on Nivolumab for metastatic malignant melanoma who developed grade III anterior uveitis. After 2 months of topical corticosteroids and cycloplegics for comfort, inflammation was resolved without stopping immunotherapy [211]. 

In other studies, ICI cessation was conducted more frequently. In Martens et al.’s review of ocular immune-related adverse events, among 134 cases of uveitis (60 anterior uveitis, eight intermediate uveitis, five posterior uveitis, 26 panuveitis, 30 VKH-like uveitis, and five undifferentiated uveitis), 124 cases required treatment with corticosteroids, either local or systemic, and 79 patients had their ICI stopped [128]. However, this study displays a greater proportion of high-grade uveitis, which could justify the increased number of patients for whom ICIs were discontinued, in comparison with the literature. The resolution of uveitis was complete in 82 cases and partial in 20 cases. Therefore, most cases responded to corticosteroid therapy and discontinuation of ICI therapy. Chaudot et al. demonstrate an association between the severity of uveitis and the incidence of ICI discontinuation, finding that 45% of anterior uveitis cases, 50% of intermediate uveitis and 71% of panuveitis cases cease their immunotherapy [72]. Similarly, in the review from Dow et al. 51.4% of patients had their ICI discontinued and 11.4% had it suspended. The majority of these cessations (83%) were due to uveitis and the rest were due to other immune-related adverse events or cancer progression [127]. 

The variability between studies in the propensity to stop ICIs demonstrates the challenge behind the decision to discontinue immunotherapy and the need to account for other factors such as the type of ICI, the type and stage of tumor, and the vital prognosis. All studies included a varying percentage of patients for whom the decision to stop the ICI was motivated by other adverse events occurring simultaneously. These aspects should also be part of the discussion between oncologists, ophthalmologists, and other specialists when ICI discontinuation is considered.

ICI discontinuation is not always favorable. In some cases, uveitis is not caused by an inflammatory reaction triggered by the ICI but is rather part of a paraneoplastic syndrome, which constitutes clinical manifestations caused by cross-reactions between antibodies against tumor antigens and self-antigens [128]. This hypothesis is supported by an intriguing case of a patient with metastatic melanoma, who was never treated with ICIs or chemotherapy, and developed VKH-like panuveitis and MAR. In this scenario, uveitis evolution could follow cancerous activity and continuation of the ICI, though it seems counter-intuitive, could potentially lead to uveitis control [128]. In these cases, continuing ICIs could help control the disease by suppressing tumor activity, thereby decreasing the inflammatory response against ocular tissue. Clinically differentiating auto-immune and paraneoplastic uveitis is difficult, and in uncertain cases or refractory cases, discontinuation of the ICI is safer 130]. In cases where ICIs are continued, the risk of uveitis relapse after finishing the steroid treatment is higher, especially if the uveitis event occurred early in the treatment course. In these cases, a longer course of steroids and progressive tapering are key strategies to avoid relapse. In some cases, continuing steroid therapy with minimal dose throughout immunotherapy is the preferred option [212,213]. 

Apart from treating ocular inflammation, a symptomatic treatment must be part of managing ICI-related uveitis. Cycloplegics are necessary to reduce pain, photophobia, and the risk of posterior synechiae. 

## 9. Management of Refractory Cases 

Local steroids are, for most cases, sufficient to induce remission of ICI-related uveitis. In cases where there is the involvement of the posterior segment, intravitreal steroid implants and peri-ocular injections can be useful and could avoid systemic steroids or ICI discontinuation which have greater impacts on tumor progression. In a study conducted by Zhou et al., all cases of ICI-related uveitis were efficiently managed with local steroid therapy and followed with imaging modalities to ensure complete remission [67]. Treatment modalities for refractory uveitis are similar to those for high-grade uveitis. Refractory cases may require initiation of systemic steroids administered either orally or intravenously, discontinuation of immunotherapy and/or initiation of immunosuppressors, immunomodulators, or biologics. Typically, the first option in refractory uveitis is systemic corticosteroids with ICI discontinuation. Only a minority of patients will require further lines of treatment with antimetabolites such as azathioprine, mycophenolate mofetil, or methotrexate. If disease control is still not achieved, biologics, such as anti-TNF can be considered but with caution. Immunomodulatory therapies are a possible alternative depending on availability [128]. 

Recurrence of ICI-related uveitis is possible and was demonstrated through many studies assessing particularly nivolumab, pembrolizumab and ipilimumab [121,214,215]. Preventing recurrence of uveitis is challenging when ICIs are continued or re-introduced. The long-term use of dexamethasone drops once uveitis control is achieved has been suggested, to maintain low ocular inflammation activity [202]. Another proposed strategy is the re-injection of corticosteroids in cases of persisting posterior segment involvement [72].

As mentioned, subclinical posterior segment inflammation can sometimes accompany anterior segment disease and go unnoticed if imaging of the posterior segment is not performed. Understanding the extent of inflammation and having the right diagnosis is essential for choosing the right therapeutic strategy. There are a few cases of patients incorrectly diagnosed with anterior uveitis based solely on clinical examination who went on to develop severe and uncontrolled panuveitis due to delay in treatment. These three patients received a topical steroid treatment initially and later developed an extension of the inflammatory activity to the posterior segment, which was refractory to systemic treatment and accompanied by a severe decrease in visual acuity reaching light perception in one case [74,216,217]. These patients had panuveitis but did not undergo posterior segment imaging such as fluorescein angiography (FA) or indocyanine green angiography (ICGA) studies. These cases highlight the importance of imaging in the detection of sub-clinical inflammation and disease extension. Establishing the right diagnosis from the beginning is crucial to orient treatment and minimize chances of refractory disease as well as the need for ICI discontinuation.

## 10. Prognosis and Complications

Most patients regain their baseline vision within one Snellen chart line with adequate topical or systemic steroid treatment [162]. In a medium-sized retrospective cohort study of 36 patients, 25 cases of uveitis were reported, among which 44% were anterior uveitis and 28% were panuveitis. Adequate treatment with local (32%), systemic corticosteroids (50%), or both (39%) allowed 81% of patients to reach either complete or partial remission. Outcomes were unavailable for three patients and one patient experienced worsening of his ocular condition [72]. Quantitatively, median final visual acuity is reported as 20/25, and was similar for anterior, posterior and panuveitis [72,127]. 

Like onset time, remission time from ICI-associated uveitis is highly variable, with some cases resolving in one week and others persisting after a year [127]. The median time to achieve uveitis control was similar when ICI was continued (25.5 days) to when ICI was discontinued (30 days); however, ICI discontinuation was more frequent with cases of severe uveitis. Similarly, in Fardeau et al.’s case series of patients treated with nivolumab and ipilimumab, at three months, outcomes were favorable with a local corticosteroid treatment regardless of whether the ICI was suspended or not. At 12 months, however, ICI continuation was associated with persistent ocular inflammatory activity, which responded well to triamcinolone sub-Tenon injections [202]. Many studies highlight the tendency of ICI-related uveitis to recur when ICIs are continued or re-introduced despite a satisfactory initial response to corticosteroids [129]. This phenomenon could be explained by an upregulation of the immune response secondary to immunotherapy which contributes to maintaining a chronic ocular inflammatory state.

Despite being the mainstay of treatment for uveitis, steroids are not without side effects. Ocular hypertension and steroid-induced glaucomatous changes are known complications that can occur with topical, intraocular, periocular, oral and intravenous administration of glucocorticoids (GC), but are more common with topical administration, responsible for 75% of cases of steroid-induced glaucoma [199,218].

Patients known for primary open-angle glaucoma are at higher risk of substantially raising their IOP; 46 to 96% of these patients experience significant IOP rise after treatment with topical GC. Newer topical steroids like loteprednol demonstrate a lower incidence of increased ocular pressure. Between 1.7 and 2.1% of patients treated with loteprednol experience a clinically significant rise in IOP [199,219,220]. 

For systemic steroids, the risk of inducing ocular hypertension is lower than for topical administration but still well reported. When used in combination, topical and systemic steroids can induce a higher and earlier IOP rise [221,222,223]. Since they are often used, either alone or in combination, ophthalmologists must stay aware of their side effects to promptly detect patients experiencing complications. 

Another well-described ocular complication of corticosteroids, especially when locally administered is the development of cataracts. In a population-based case-control study of 15,479 people in England, the odds ratio of developing cataracts with exposure to ocular and systemic corticosteroids were 2.12% and 1.59%, respectively [224]. Among 914 patients, that is 1192 eyes, treated with corticosteroids periocular injections, the incidence of the development of cataracts was 20% [225].

## 11. Monitoring and Follow-Up

Monitoring and follow-up depend on the severity of uveitis and the treatment modality. The median treatment time for topical corticosteroids is 11 weeks (8 weeks to 10 months) and for oral corticosteroids, 10 weeks (4 weeks to 6 months). In a retrospective case series conducted in Northern California on 5061 patients treated with ICIs, in which 31 cases of ocular inflammation were identified, the average follow-up time was 16 ± 18 months, ranging from 0 to 71 weeks [147]. Recurrence of uveitis is possible, and even likely when ICIs are continued or re-introduced. Therefore, it is important to follow up even after the resolution of the current episode to ensure complete remission or intervention if needed.

### Neoplastic Outcome

As previously explained, immunotherapy and therapeutic strategies for uveitis control have opposite mechanisms of action. Many studies raise the concern about the effects that uveitis treatment can have on tumor progression. These effects seem to be minimized when treatment is limited to local corticosteroid therapy, but the effect of systemic corticosteroids, ICI discontinuation, immunosuppressants and biologics have not extensively been studied yet.

In a retrospective case series of 28 patients treated with either local or systemic steroids or a combination of both, nine patients (32%) had stable tumor activity, ten patients (36%) were in remission from their cancer and among these patients, seven attained complete remission, of which four patients had their ICIs discontinued. Tumor progression was reported in four patients (14%); all four patients had discontinued their ICI and two of them received systemic steroids. Neoplastic outcome was unavailable for five patients (18%) [72]. A recent literature review conducted by Chaudot et al. reported 21 cases of ICI-related uveitis in which cancer progression was noted, 26 cases of partial or complete remission and eight cases of stable cancerous activity. For most cases, however, (66 patients) the data on neoplastic outcome was unavailable [72].

## 12. Current Research and Future Directions

### 12.1. Summary of Recent Studies

In summary, recent studies agree on the importance of individualizing treatment for the optimal management of uveitis associated with ICI. Most cases of mild anterior uveitis can be managed with topical steroids, but imaging should be performed to rule out any inflammatory process in the posterior segment. Local corticosteroids are preferred over systemic corticosteroids for anterior segment involvement; for posterior segment disease, peri-ocular or intravitreal injections can be used. Treatment should be initiated as soon as possible to control the disease without needing to discontinue immunotherapy. The decision to discontinue ICI should be discussed with oncologists. In some patients, reintroduction of ICIs is possible, but for those exhibiting grade 4 uveitis, it is not recommended. If reintroduction of ICIs is attempted, a daily low dose of corticosteroids is recommended to prevent a relapse. More severe or refractory anterior uveitis, intermediate, and posterior uveitis usually require systemic corticosteroids in addition to topical steroids and possibly discontinuation of the ICI. Immunosuppressors or biologics in patients experiencing adverse events to glucocorticoids is an alternative that should be used with caution in the context of unknown neoplastic outcomes. Some studies highlighted the importance of confirming the diagnosis of uveitis through imaging with FA and ICGA to help clinicians detect sub-clinical inflammation, specify the diagnosis, follow the evolution, and tailor treatment. 

### 12.2. Future Therapeutic Strategies and Research Needs

In terms of future therapeutic strategies, alternative biologic therapy could be further studied to better identify targeted molecules for the treatment of ICI uveitis. In a retrospective, nonrandomized clinical study conducted by Utz et al., a significant number of children suffering from chronic anterior uveitis refractory to conventional TNF-alpha therapy were evaluated. Among 52 children, nine were refractory to TNF-alpha and methotrexate and received alternative biologics such as abatacept, tocilizumab and golimumab. All nine children experienced remission and had their glucocorticoid therapy tapered to a dose of two drops per day without causing any complications [226]. These biological agents seem efficient in the treatment of chronic anterior uveitis in children but need to be tested for ICI-related uveitis in the context of patients treated for cancer. Future research should aim to expand the inventory of molecules available for the treatment of ICI-related uveitis.

A major issue in the management of severe ocular adverse events is the need to treat ocular problems and maintain ocular health without interfering with immunotherapy. As described previously, corticosteroids, immunosuppressors and biologics can alter the efficacy of ICIs and have ocular side effects. Research is now focused on the development of therapeutic strategies that can efficiently treat ocular adverse events, like uveitis, without interfering with cancer treatments. Researchers are currently studying bacterial-derived enzymes that can inactivate auto-antibodies by cleaving or by deglycosylating immunoglobulins [227,228]. Another class of molecules, inhibitors of the neonatal Fc receptors, are also currently studied. The neonatal Fc receptors are necessary to prevent IgG degradation. In patients diagnosed with myasthenia gravis, a phase II trial demonstrated the reduction of IgG concentrations following treatment with efgartigimod and rozanolixizumab, inhibitors of the neonatal Fc receptor [229,230]. 

## 13. Recommendations for Clinical Practice and Research

(1)A baseline ophthalmology examination before initiation of ICI is recommended for patients with risk factors for ICI-induced uveitis. These include prior episodes of uveitis or other ocular inflammatory conditions, a history of ocular trauma or surgery, and patients with a medical history of autoimmune disease or renal failure. Indeed, 27% of patients with prior autoimmune diseases develop IRAEs following ICI initiation and up to 51% of patients with a prior history of uveitis develop ICI-induced uveitis [74,231,232,233]. Routine screening of at-risk populations would also facilitate the diagnosis and treatment of pre-existent diseases before the initiation of immunotherapy;(2)Clinical presentations of ICI-induced uveitis vary and include anterior, intermediate, posterior, VKH-like, and BU-like uveitis. When a patient undergoing ICI treatment presents with symptoms of ocular inflammation and reduced vision, clinical suspicions of ICI-induced uveitis must be high and need to be confirmed via a complete uveitis workup;(3)ICI-induced uveitis can present as panuveitis. Therefore, patients presenting with signs of anterior uveitis and reduced vision should undergo further imaging (e.g., OCT) to adequately rule out any posterior segment involvement. Thereby, the correct treatment modality can be initiated, leading to the preservation of visual prognosis;(4)Our treatment recommendations align with most studies in the literature. Mild anterior uveitis should be treated with topical steroids. Moderate to severe anterior, intermediate, and posterior uveitis should be treated with intravitreal, periocular, or topical steroids depending on the segment involved, along with systemic corticosteroids. ICI therapy should also be suspended following a discussion with the oncologist. We also recommend avoiding re-initiation of ICI therapy in grade 4 uveitis. Once uveitis is resolved, corticosteroids should be progressively tapered to avoid relapses of ocular inflammation;(5)Since uveitis can occur at any time along the treatment course, interdisciplinary collaboration between ophthalmologists and oncologists is crucial to allow earlier consultations in ophthalmology and to preserve visual prognosis. Oncology follow-up is frequent and represents good opportunities to screen for ocular symptoms. A standard questionnaire may be provided to the oncology team and the patient to facilitate the recognition of ophthalmic red flags. Decisions relative to the treatment of ocular adverse events should be made by consensus in a multidisciplinary meeting to ensure an optimized and individualized therapeutic strategy [202];(6)Ophthalmologists should stay aware of ocular metastatic disease. Despite the eye’s immune privilege, metastatic disease can reach ocular and periocular tissues. Several cases of masquerade uveitis later found to be ocular cancer metastases were reported. A key clinical distinction is the often bilateral and symmetric nature of uveitis [127]. However, there can be exceptions. Manusow et al. describe a case of a 36-year-old female treated with pembrolizumab for metastatic cutaneous melanoma. After complete remission, she developed bilateral vitreous metastasis without choroidal involvement, associated with retinal vasculitis detected through angiography. Retinal vasculitis is believed to be a paraneoplastic phenomenon to retinal cell antigens since it completely resolves immediately after performing a vitrectomy. Examination findings led clinicians to suspect metastatic disease and perform a vitreous biopsy. On examination, vitreous cells were atypical, adherent in small clusters without any signs of damage to the vitreous gel [64]. In a second case report, a 63-year-old woman treated with ipilimumab and prolonged high-dose steroids for metastatic malignant carcinoma presented with signs and symptoms of right panuveitis. Large vitreous opacities were present bilaterally on dilated fundus examination, with pale yellow retinal lesions. Clinicians suspected candida endophthalmitis and performed pars plana vitrectomy, which revealed metastatic disease [234]. Despite a partial lung response to immunotherapy, the vitreous and retina, considered immunologically privileged tissues, demonstrated metastatic progression.

## 14. Conclusions

In conclusion, among ocular IRAEs, uveitis post-ICI is frequent and can occur early on during the treatment course. The specific mechanism of ICI-induced uveitis is unknown, but a combination of factors—including cross-reactivity with cancer antigens and the microbiome—likely plays a role. Melanoma and lung cancers are the most associated with ICI uveitis. ICI-induced uveitis can have various clinical presentations ranging from mild anterior uveitis to severe sight-threatening panuveitis. Although rare, VKH-like uveitis and birdshot-like uveitis are reported following ICI therapy. A variety of imaging modalities are at the clinician’s disposition, and it is important to characterize the extent of intraocular inflammation to determine treatment options and prognosis. In terms of treatment, ICI-induced uveitis requires prompt targeted topical and/or systemic corticosteroids, with the possible addition of immunosuppressants or biologics depending on the severity and the evolution. Care must be taken when starting these medications because of their immunocompromising effect and side effects. Collaboration between ophthalmologists and oncologists is crucial to adequately manage oIRAEs and reduce the risk of tumor progression. Research focusing on understanding the mechanisms and finding new therapies targeted for ICI-induced uveitis that will not interfere with cancer treatment is needed.

## Figures and Tables

**Figure 1 diagnostics-14-00336-f001:**
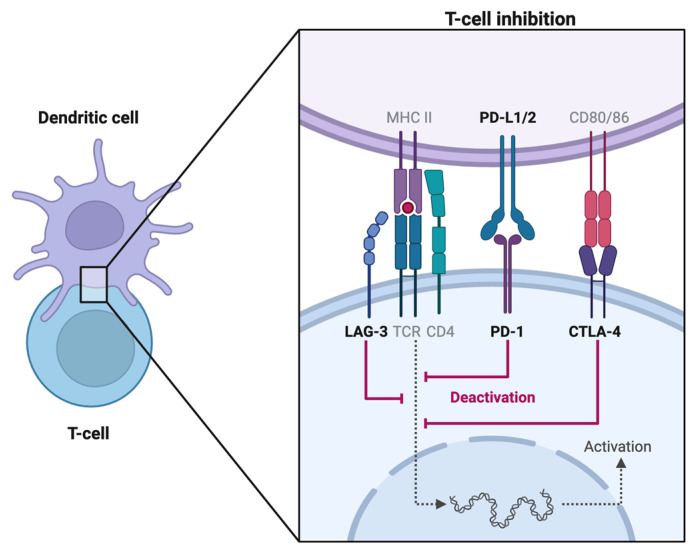
Interactions of inhibitory checkpoint molecules. Adapted from “T-cell Deactivation vs. Activation”, by BioRender.com. Retrieved from https://app.biorender.com/biorender-templates (accessed 3 November 2023).

**Figure 2 diagnostics-14-00336-f002:**
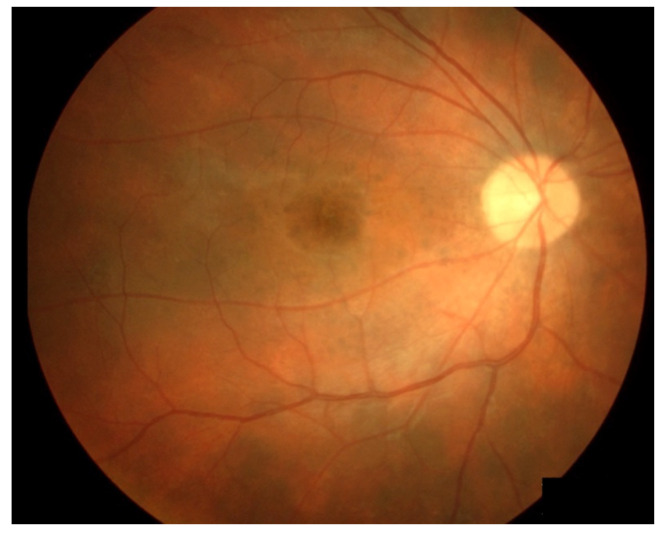
Fundus photographs of a VKH patient who developed “sunset glow fundus”. From Hirooka et al., 2017 [133].

**Figure 3 diagnostics-14-00336-f003:**
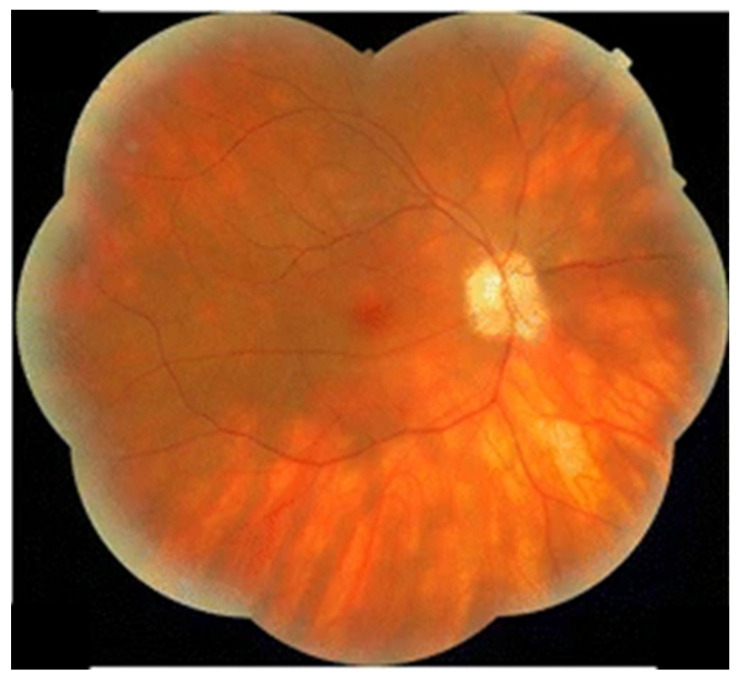
Birdshot fundus lesions. From Minos et al., 2016 [156].

**Table 1 diagnostics-14-00336-t001:** Non-ocular adverse events associated with ICI therapy.

Organ System	Reported Adverse Events
**Cardiovascular**	Arrhythmia [21] **Cardiac failure** [22] Hypertension [23] Myocardial infarction [22] Myocarditis [24,25] **Stroke** [22]
**Endocrine**	**Hypothyroidism**, hyperthyroidism [26,27] **Hypophysitis** [28,29] **Hyperglycemia** [30] **Loss of appetite** [26,31]
**Gastrointestinal/renal**	**Colitis/Diarrhea** [32] Constipation [26] Esophageal achalasia [33] **Hepatitis** [34] **Nausea** [26] Nephritis [35] Pancreatitis [26]
**General**	**Fatigue** [31] **Pyrexia** [31]
**Musculoskeletal**	Inflammatory myopathy [36] Arthropathy [37]
**Nervous**	Encephalopathy [20] Facial palsy [38] Hearing loss, vertigo [39,40] Neuropathy [23]
**Dermatological**	Alopecia [23] **Dermatitis** [14] **Rash** [26] Pruritus [26] Photosensitivity [26] Vitiligo [26,41]
**Respiratory**	Cough [31] Dyspnea [31] **Pneumonitis** [26]
**Other**	Sarcoidosis [42,43] Hematological disturbances

Note: common side effects (those affecting between 1 in 10 and 1 in 100 people) are **in bold** [14,20].

**Table 2 diagnostics-14-00336-t002:** Grading of Uveitis According to the CTCAE Criteria.

Grade 1	Grade 2	Grade 3	Grade 4
Anterior uveitis with trace cells	Anterior uveitis with 1+ or 2+ cells	Anterior uveitis with 3+ or greater cells; intermediate posterior or pan-uveitis	Best corrected visual acuity of 20/200 or worse in the affected eye

**Table 3 diagnostics-14-00336-t003:** Naranjo criteria for determining if an adverse event is related to drug administration.

Question		Yes	No
1. Are there previous conclusive reports on this reaction?		+1	0
2. Did the adverse event appear after the suspected drug was administered?		+2	−1
3. Did the adverse event improve when the drug was discontinued, or a specific antagonist was administered?		+1	0
4. Did the adverse event reappear when the drug was readministered?		+2	−1
5. Are there alternative causes that could on their own have caused the reaction?		−1	+2
6. Did the reaction reappear when a placebo was given?		−1	+1
7. Was the drug detected in blood or other fluids in concentrations known to be toxic?		+1	0
8. Was the reaction more severe when the dose was increased or less severe when the dose was decreased?		+1	0
9. Did the patient have a similar reaction to the same or similar drugs in any previous exposure?		+1	0
10. Was the adverse event confirmed by any objective evidence?		+1	0
Total score	Interpretation of Scores		
≥9	Definite. The reaction (1) followed a reasonable temporal sequence after a drug or in which a toxic drug level had been established in body fluids or tissues, (2) followed a recognized response to the suspected drug, and (3) was confirmed by improvement on withdrawing the drug and reappeared on re-exposure.		
5 to 8	Probable. The reaction (1) followed a reasonable temporal sequence after a drug, (2) followed a recognized response to the suspected drug, (3) was confirmed by withdrawal but not by exposure to the drug, and (4) could not be reasonably explained by the known characteristics of the patient’s clinical state.		
1 to 4	Possible. The reaction (1) followed a temporal sequence after a drug, (2) possibly followed a recognized pattern to the suspected drug, and (3) could be explained by characteristics of the patient’s disease.		
≤0	Doubtful. The reaction was likely related to factors other than a drug.		

From: [169].

**Table 4 diagnostics-14-00336-t004:** Treatment recommendations for ocular adverse events associated with ICIs.

	ASCO 2018	STIC 2017	NCCN 2019
Grade 1	Continue ICI	Continue ICI	Continue ICI
	Artificial tears		Artificial tears
Grade 2	Suspend ICI	Suspend ICI	Suspend ICPI
	Systemic corticosteroids	PO prednisone	Prednisone or methylprednisolone
	Resume ICI once off systemic corticosteroids (SC), or if on SC for other organ maintenance at less than 10 mg/day	PO prednisone 0.5–1 mg/kg/day or IV methylprednisolone 0.5–1 mg/kg/day. If no improvement, increase dose to 2 mg/kg/day. If improved to Grade 1 or less, taper over 4–6 weeks.	Resume ICPI once symptoms are Grade 1 or less without corticosteroids
	Topical corticosteroids, cycloplegic agents		Local corticosteroids
Grade 3	Suspend ICPI	Suspend ICPI, permanently discontinue if symptoms do not improve over 4–6 weeks	Permanently discontinue ICPI
	Prednisone 1–2 mg/kg/day or methylprednisolone 1–2 mg/kg/day, tapered over 4–6 weeks	Prednisone 1–2 mg/kg/day or equivalent dose of methylprednisolone. If no improvement over 2–3 days, add additional immunosuppressant. If improved to Grade 1 or less, taper over 4–6 weeks.	Prednisone or methylprednisolone
			Local corticosteroids
Grade 4	Permanent discontinuation of ICPI	Permanent discontinuation of immunotherapy	Permanently discontinue ICPI
	IV prednisone 1–2 mg/kg/day or methylprednisolone 0.8–1.6 mg/kg	Prednisone 1–2 mg/kg/day or equivalent dose of methylprednisolone. If no improvement over 2–3 days, add additional immunosuppressant.	Prednisone or methylprednisolone
	Intravitreal, periocular, or topical corticosteroids		

ASCO: American Society of Clinical Oncology; SITC: Society for Immunotherapy of Cancer; NCCN: National Comprehensive Cancer Network.

## Data Availability

Not applicable.

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
