# Peer review of "Diagnosing and Managing Uveitis Associated with Immune Checkpoint Inhibitors: A Review"

_diagnostics, 2024, doi:10.3390/diagnostics14030336_

Round 1

Reviewer 1 Report

Comments and Suggestions for Authors

In their article, the authors give a comprehensive review on uveitis caused by immune checkpoints inhibitors give for cancer treatment. Some aspects should be addressed before publication

Table 1, the legend states that common site effects are depicted in bold, but none are actually depicted in bold. In addition, for all tables, please define “common”

Lines 171 to 176, in the retrospective analysis mentioned, were these data selected for specific drugs, dosing or duration of treatment? Please elaborate.

Line 213, 214 “has failed in multiple clinical trials”. I do not understand, why is a drug given that has failed in clinical trials? Please explain.

Line 331 ff, the authors state that the main uveitis is anterior with 37,7 % followed by pan with 34 % and lest is intermediate with 0.01%. While this is of course mathematically true, the difference between 37.7 and 34 is small, so is this difference relevant? And even more, 0.01% is very little, is this number correct? If so, this strong discrepancy between anterior and pan on the one and intermediate on the other should be discussed.

Line 341/342, “compared to other ICIs”, there are not so many more than those mentioned, if I understood the introduction right (LAG-3 and PD-L1), or did this study include more ICI?

Line 364/365, “more so than other drugs”, what other drugs induce VKH?

Line 381 “BU … 6%-8% of posterior uveitis cases”, of all posterior uveitis cases or only ICI-induces?

Line 397 “left > right eye”, please phrase what is meant to say

Line 389, are there really only two cases of ICI-induced BU?

Line 433 “Birdshot-like uveitis preferentially affects older Caucasians”, if only two cases are known, this claim cannot be made. Please include reference and, if they exist, more studies on BU ICI.

Line 500 “pallidum” has to be written in italics

Line 533 ff, it is not clear if and how the B-scan ultrasonography is applied in uveitis. Please explain or delete

Line 585 ff, it is not clear of these percentages are meant of all patients or of patients with uveitis

Line 618, how is “uncontrolled” and “serious” disease defined? What are the differences?

Line 615 ff and lines 645 ff, it is not clear why these studies are discussed in different sections. If there is a reason, please explain, otherwise, these sections should be combinded

Line 696, there is an additional space before “Chronic”

Line 698, there is a “.” missing after the reference

Line 712, there is a space missing between “10” and “mg”

Line 874, the authors claim that it is not clear whether the occurrence of ICI is correlated to a better survival, but on page 3, lines 109 ff, there are clear statements about studies showing that it is indeed correlated with better survival

Line 986, the authors explicitly mention renal cancer in the conclusion, but renal cancer is only mentioned in one sentence (line 170) in the associations with different cancer types. A more detailed coverage of ICI-induced uveitis in renal cancers in section “Association with different cancer types” would be warranted if renal cancers are to make it into the conclusions.

Reviewer 2 Report

Comments and Suggestions for Authors

I applaud the authors for this useful and informative review. My only concern is that the article suggests providing a comprehensive overview of drug-induced uveitis in the context of ICI therapy, but it lacks a methods section. A comprehensive review should include a methods section detailing the search strategies for relevant articles and also a section detailing the criteria used for choosing the most relevant and reliable articles. If the authors have not followed this methodology, they should not label their review as comprehensive.

There are some minor typos in the article that should be corrected. For example, on page 3: “In a systematic review comparing 103 ipilimumab 3 mg/kg and 10 kg/mg, authors found that the higher dosage group had a 3.10 104 greater chance of developing high-grade IRAEs [14].”

"10 kg/mg" should be corrected to "10 mg/kg."

Comments on the Quality of English Language

Needs minor edditing.
